# Phase separation of a plant virus movement protein and cellular factors support virus-host interactions

Shelby L. Brown, Dana J. Garrison, Jared P. May*

Department of Cell and Molecular Biology and Biochemistry, School of Biological and Chemical Sciences, University of Missouri-Kansas City, Kansas City, Missouri, United States of America

* jpmay@umkc.edu

## Abstract

Both cellular and viral proteins can undergo phase separation and form membraneless compartments that concentrate biomolecules. The p26 movement protein from single-stranded, positive-sense *Pea enation mosaic virus 2* (PEMV2) separates into a dense phase in nucleoli where p26 and related orthologues must interact with fibrillarin (Fib2) as a pre-requisite for systemic virus movement. Using *in vitro* assays, viral ribonucleoprotein complexes containing p26, Fib2, and PEMV2 genomic RNAs formed droplets that may provide the basis for self-assembly *in planta*. Mutating basic p26 residues (R/K-G) blocked droplet formation and partitioning into Fib2 droplets or the nucleolus and prevented systemic movement of a Tobacco mosaic virus (TMV) vector in *Nicotiana benthamiana*. Mutating acidic residues (D/E-G) reduced droplet formation *in vitro*, increased nucleolar retention 6.5-fold, and prevented systemic movement of TMV, thus demonstrating that p26 requires electrostatic interactions for droplet formation and charged residues are critical for nucleolar trafficking and virus movement. p26 readily partitioned into stress granules (SGs), which are membraneless compartments that assemble by clustering of the RNA binding protein G3BP following stress. G3BP is upregulated during PEMV2 infection and over-expression of G3BP restricted PEMV2 RNA accumulation >20-fold. Deletion of the NTF2 domain that is required for G3BP condensation restored PEMV2 RNA accumulation >4-fold, demonstrating that phase separation enhances G3BP antiviral activity. These results indicate that p26 partitions into membraneless compartments with either proviral (Fib2) or antiviral (G3BP) factors.

## Author summary

We demonstrate that the p26 movement protein from *Pea enation mosaic virus 2* partitions with cellular proteins fibrillarin and G3BP. Using *in vitro* assays, we determined that basic residues and electrostatic interactions are critical for p26 phase separation and partitioning in pre-formed fibrillarin droplets. Furthermore, mutation of either basic or acidic residues disrupted nucleolar trafficking of p26 and prevented the rescue of a movement-

**Data Availability Statement:** All relevant data are within the manuscript and its Supporting Information files.

**Funding:** This work was supported by University of Missouri-Kansas City (UMKC) institutional start-up

funds to J.P.M. The funders had no role in study design, data collection and analysis, decision to publish, or preparation of the manuscript.

**Competing interests:** The authors have declared that no competing interests exist.

deficient TMV vector in *Nicotiana benthamiana*. Overexpression of the stress granule nucleator G3BP severely restricts PEMV2 RNA accumulation. However, deletion of the NTF2 domain required for G3BP condensation alleviated the antiviral activity towards PEMV2.

This study expands our knowledge of virus-host interactions occurring through phase transitions particularly regarding plant viruses where limited evidence currently exists.

## Introduction

Cellular organelles are membrane-bound compartments that are critical for eukaryotic cell function. RNA viruses frequently co-opt organelles to promote virus replication, including the endoplasmic reticulum (ER) [1], mitochondria [2], nucleus [3], and Golgi apparatus [4]. Much attention has been recently directed towards membraneless organelles that form through phase separation, which transforms a single-phase solution into a dilute phase and dense phase that concentrates biomolecules, such as proteins or RNAs [5, 6]. Proteins that undergo phase separation contain intrinsically disordered regions (IDRs) that self-associate to form oligomers [7]. Many IDR-containing proteins have RNA-recognition motifs that non-specifically bind RNA and fine-tune phase separation by controlling material exchange, shape, and rigidity of liquid droplets [7, 8]. Proteins that phase separate are often enriched in arginine residues that promote phase separation through cation-pi interactions with aromatic contacts [9]. In addition, hydrophobic interactions can stabilize phase separations of low-complexity domains [10].

Membraneless organelles exist as liquids, gels, or solids [11]. The most notable examples of cellular membraneless organelles are the nucleolus and cytoplasmic P-bodies [12]. Less dynamic stress granules (SGs) also form in the cytoplasm and allow host cells to repress translation and influence messenger RNA (mRNA) stability in response to various stresses [13]. SGs are visible by microscopy within minutes following stress and contain the RNA binding protein G3BP1 that self-associates to induce SG formation [14]. SGs contain a stable inner core and an outer shell that is formed by weak electrostatic and/or hydrophobic interactions [15]. The G3BP1 inner core has limited circularity and is resistant to dilution (both atypical for liquid droplets) and is surrounded by a highly dynamic shell structure [15, 16]. Interestingly, G3BP1 can have either proviral [17–19] or antiviral roles [20–22] in RNA virus infection cycles.

Phase separation of viral proteins has largely been associated with negative-sense RNA viruses from the *Mononegavirales* family that form membraneless virus factories [23, 24] such as Negri bodies that form during Rabies virus infections [25–27]. In contrast, many positive-sense RNA viruses, including members of the *Tombusviridae* family, form membranous replication organelles that concentrate virus replication complexes [28, 29]. Recent work has demonstrated that the nucleocapsid (N) protein from the positive-sense SARS-CoV-2 coronavirus undergoes phase separation stimulated by the 5' end of its genomic RNA [30, 31]. N protein partitions into droplets of heterogeneous nuclear ribonucleoproteins like TDP-43, FUS, and hnRNPA2 [32] and also interacts with G3BP1, which attenuates SG formation [33, 34]. The role of N droplet formation in the virus infection cycle is unclear but could be involved in nucleocapsid assembly and genome processing [35]. Although limited evidence for phase separation of plant virus proteins exists [36], a recent study demonstrated that *Turnip mosaic virus* inhibits the formation of phase-separated nuclear dicing bodies (D-bodies) that are responsible for microRNA processing and antiviral defense [37, 38]. While these findings demonstrate cellular condensates can possess antiviral activity, examples of plant virus proteins that use phase separation to support virus-host interactions have not been reported.

*Pea enation mosaic virus 2* (PEMV2) is a small (4,252 nt), positive-sense RNA plant virus belonging to the umbravirus genus in the *Tombusviridae* family. The PEMV2 long-distance movement protein p26 is required for trafficking viral RNA through the vascular system of infected plants [39]. Both p26 and the umbravirus *Groundnut rosette virus* (GRV) orthologue (pORF3) form large cytoplasmic granules [40–42], but also target cajal bodies in the nucleus and eventually partition in the nucleolus [43–45]. Umbravirus p26 orthologues interact with nucleolar fibrillarin, which is required for long-distance movement of the viral genomic RNA [40, 44, 45]. In addition to umbraviruses, polerovirus *Potato leafroll virus* (PLRV) and the satellite RNA of potexvirus *Bamboo mosaic virus* (satBaMV) encode proteins that also localize to the nucleolus and interact with fibrillarin to support systemic movement [46–48]. Fibrillarin forms droplets that make up the dense fibrillar component (DFC) of the nucleolus, which shares a similar structure with SGs [15, 49]. Although the nucleolus itself is a phase separation and these plant virus proteins interact directly with fibrillarin, the role of viral protein phase separation in the infection cycle has not been investigated.

In this report, we demonstrate that PEMV2 p26 assembles viscous condensates *in vivo* with low intra-droplet diffusion (i.e., droplets are poorly dynamic). Viral ribonucleoprotein (vRNP) complexes containing p26, fibrillarin, and PEMV2 genomic RNAs were reconstituted *in vitro* through phase separation, which we hypothesize recapitulates the *in vivo* event necessary for systemic trafficking. Charged residues played critical roles in p26 droplet formation, nucleolar localization, and movement of a TMV vector, suggesting that phase separation and virus movement are connected. Finally, p26 partitions into SGs, and G3BP over-expression restricts PEMV2 RNA accumulation >20-fold. A G3BP mutant incapable of forming a dense phase only reduced PEMV2 RNA accumulation by 5-fold, suggesting that phase separation is involved in the antiviral host response.

## Results

### p26 forms poorly dynamic condensates *in vivo*

To visualize and characterize the material properties of p26 granules *in vivo*, green fluorescent protein (GFP) was fused to the C-terminus of full-length p26 and expressed from the Cauliflower mosaic virus (CaMV) 35S promoter (Fig 1A) following agroinfiltration of *Nicotiana benthamiana* leaves. To maximize transient gene expression, the p14 RNA silencing suppressor from *Pothos latent virus* [50] was included in all infiltrations for this study. GFP expressed from the CaMV 35S promoter failed to form granules and was evenly distributed throughout the cytoplasm and nucleus of the cell (i.e., outside of the large vacuole that comprises most of the cellular space) (Fig 1B, Left). In contrast, p26$_{WT}$ formed large cytoplasmic granules (Fig 1B, Right). To probe intra-droplet dynamics, we used fluorescence recovery after photobleaching (FRAP) assays [51]. If p26 granules are highly dynamic liquid droplets, then FRAP recovery should be rapid and complete. Conversely, if p26 forms solid aggregates, no fluorescence recovery was expected. p26$_{WT}$ granules recovered nearly 50% by 30 seconds post-bleaching (Fig 1C), indicating that these p26 droplets had measurable fluidity. However, since p26$_{WT}$ failed to fully recover, p26 appears to form viscous condensates *in vivo* with poor intra-droplet dynamics.

### p26 contains an intrinsically disordered region (IDR) and phase separates via electrostatic interactions

Since IDRs typically drive phase separation [52], we subjected p26 to the IDR prediction program IUPred [53], which identified an arginine-rich disordered region spanning amino acids

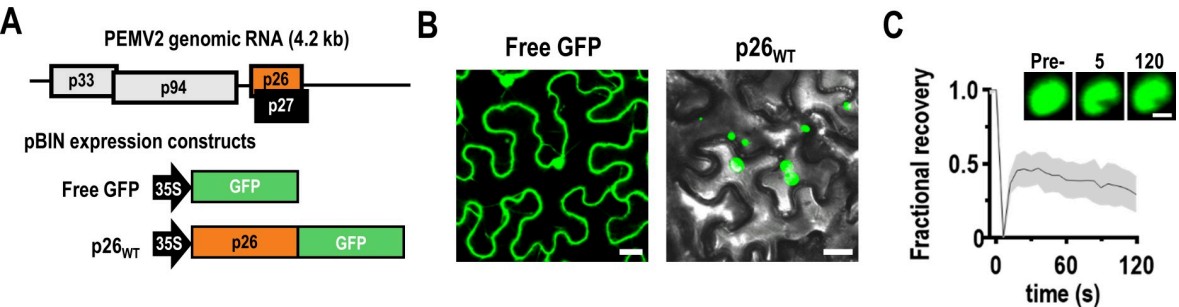

**Fig 1. p26 forms poorly dynamic condensates *in vivo*.** (A) Genomic organization of the single-stranded positive-sense RNA genome of PEMV2. Free GFP and p26 C-terminally fused with GFP (p26_WT) were expressed from binary expression plasmids under the constitutive CaMV 35S promoter. (B) Free GFP or p26_WT were agroinfiltrated alongside the p14 RNA silencing suppressor in *N. benthamiana* and imaged at 2 dpi using confocal microscopy (488 nm). Note that the majority of plant mesophyll cells is taken up by a single large vacuole. Differential interference contrast (DIC) microscopy was used for p26_WT to visualize cell borders. Bar scale: 20 μm. (C) FRAP analyses were performed by photobleaching cytoplasmic condensates and monitoring fluorescence recovery at 5 s intervals. A representative p26_WT condensate is shown before photobleaching, immediately following photobleaching (5 s), and at 120 s. Bar scale 5 μm. Average FRAP intensity is shown from seven FRAP experiments and shaded area represents standard deviations.

1–132 (Fig 2A and 2B). The same region was also predicted to have the highest propensity to phase separate using the catGRANULE algorithm (Fig 2C) [54]. To determine if the identified p26 IDR drives droplet formation, IDR_WT or amino acids 133–226 (C-term) along with N-terminal histidine tags or the tag alone was fused to the N-terminus of GFP, and proteins purified after expression in *E. coli.* were assessed for size and purity by SDS-PAGE (Fig 2D). To assay for induction of phase separation, proteins were combined with 10% PEG-8000 and droplet formation was observed by confocal microscopy and by measuring solution turbidity (OD_600). As expected, IDR_WT assembled into condensed droplets as observed by both confocal microscopy (Fig 2E) and turbidity assays (Fig 2F). In contrast, both free GFP and C-term remained in a single phase (Fig 2E and 2F). Although the N-terminal histidine tags did not influence IDR_WT phase separation propensity, particle size, or resistance to 1,6-hexanediol that selectively dissolves liquid condensates [55] (S1 Fig), FRAP recovery of IDR_WT dramatically increased following His-tag removal suggesting that the histidine tracts influenced droplet dynamics *in vitro* (S1 Fig).

Electrostatic interactions that drive self-assembly and droplet formation can be inhibited by high salt concentrations [56]. Confocal microscopy revealed significantly reduced phase separation of IDR_WT in the presence of 1 M NaCl (Fig 2E), which was confirmed by comparing total droplet areas (%) from three representative 20x fields for each condition (Fig 2G). These results support p26 assembling in a dense phase through electrostatic interactions. To determine the saturation concentration ($C_{sat}$) of IDR_WT and further investigate NaCl-mediated reduction in droplet formation, a phase diagram was generated using confocal microscopy. The apparent $C_{sat}$ was 2 μM and IDR_WT was sensitive to NaCl in a dose-responsive manner as 600 mM and 800 mM NaCl blocked droplet formation of 2 μM and 4 μM IDR_WT, respectively (Fig 2H).

Deletion of IDR sequence 5'-RRRARR-3' (amino acids 100–105), which comprises a conserved nuclear localization signal (NLS) [57] and 16% of the basic residues within the IDR, did not affect droplet formation (Fig 2E–2G), demonstrating that the NLS is not required for phase separation. However, when all basic or acidic residues in IDR_WT were mutated to glycine (IDR_R/K-G or IDR_D/E-G, respectively), IDR_R/K-G failed to phase separate while IDR_D/E-G showed significantly reduced phase separation compared to IDR_WT when examined by turbidity assays (Fig 2F), total droplet area (Fig 2G), and mean condensate size (Fig 2I). At higher

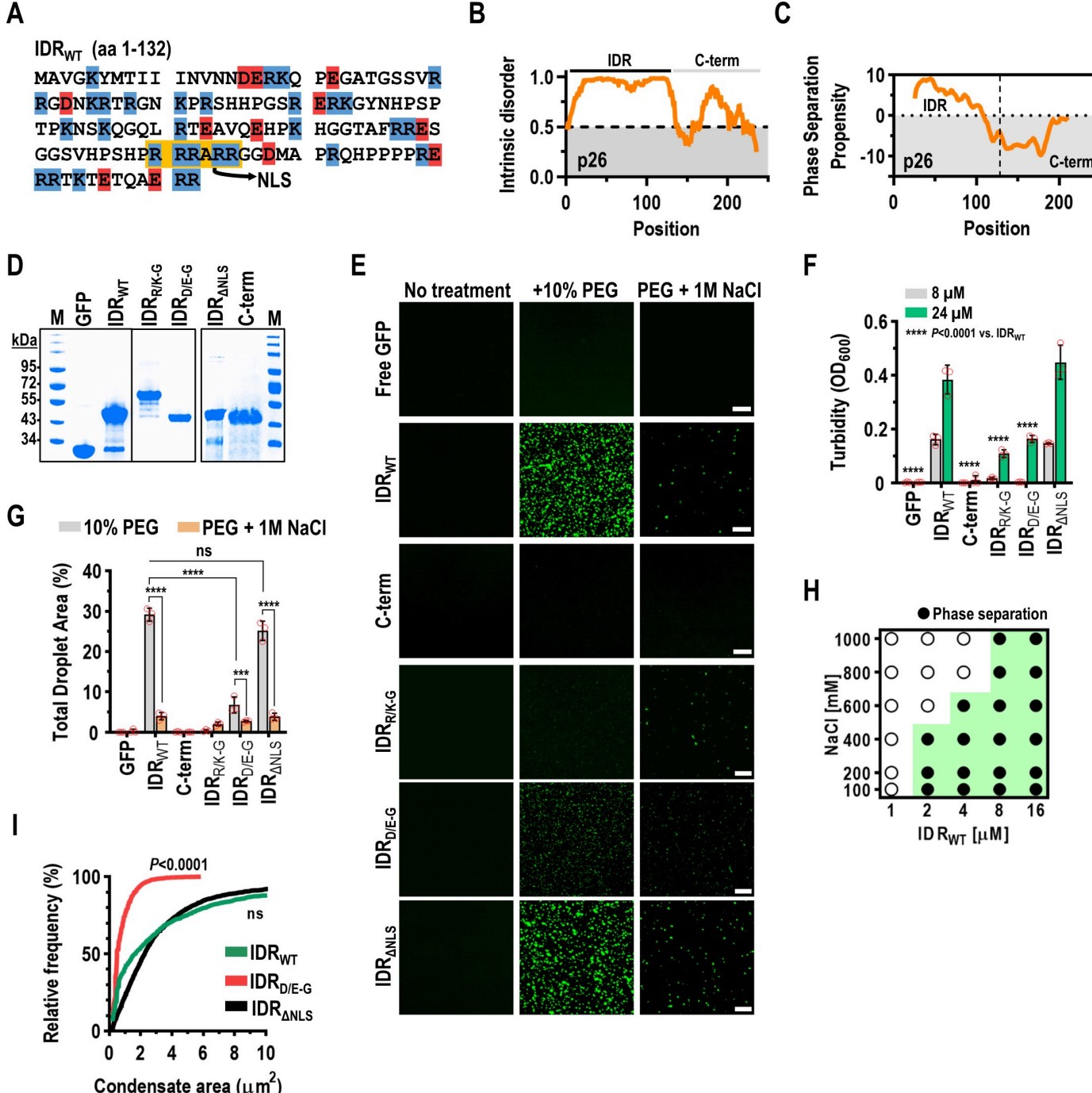

**Fig 2. p26 is intrinsically disordered and phase separates through electrostatic interactions.** (A) The p26 IDR (amino acids 1–132) is shown with highlighted residues corresponding to basic (blue) or acidic (red) residues. The conserved nuclear localization signal (NLS) is highlighted in yellow. (B) Graphical representation of predicted intrinsic disorder in p26 using IUPRED [53]. (C) Graphical representation of predicted phase separation propensity within p26 using the catGRANULE algorithm [54]. (D) N-terminal His-tagged recombinant proteins were analyzed by SDS-PAGE to assess size and purity. Proteins were stained using Coomassie Blue. Marker (M) sizes are shown in kilodaltons (kDa). $IDR_{R/K-G}$ migrated more slowly than expected both *in vitro* and *in vivo* (see Fig 6B). (E) *In vitro* droplet formation was visualized by confocal microscopy. Eight micromolar protein was used for all assays and 10% PEG-8000 was added as a crowding agent (Middle panels). One molar NaCl was added to disrupt electrostatic interactions (Right panel). Bar scale: 20 μm. Images in all panels are representative of at least two independent experiments. (F) Turbidity assays ($OD_{600}$) using either 8 μM or 24 μM protein were performed for all constructs. **** $P < 0.0001$ by two-way ANOVA with Dunnett's multiple comparisons test vs. $IDR_{WT}$. Error bars denote standard deviations and individual data points (red circles) represent three biological replicates. (G) Total droplet areas (%) were measured from confocal images using ImageJ. Error bars denote standard deviations and red circles represent three 20x fields for each assay. *** $P < 0.001$, **** $P < 0.0001$, ns: not significant using two-way ANOVA with Sidak's multiple comparisons test. (H) Phase diagram

for IDR$_{WT}$ over a range of protein and NaCl concentrations. Results are representative of two independent experiments. (I) Mean condensate sizes for IDR mutants (excluding IDR$_{R/K-G}$) were plotted by cumulative distribution frequency. Particle sizes were measured from three representative 20x fields using ImageJ. $P$ values represent results from two-tailed Mann-Whitney tests compared to IDR$_{WT}$. ns: not significant.

concentrations (24 μM), IDR$_{R/K-G}$ formed non-uniform aggregates with significantly reduced circularity compared to IDR$_{WT}$ and IDR$_{D/E-G}$ droplets (S2 Fig). When all potential cation-pi or hydrophobic interactions were disrupted by mutating all arginines to lysines (IDR$_{R-K}$) or all hydrophobic residues to serine (IDR$_{VLIMFYW-S}$), respectively, total droplet areas, turbidities, and mean condensate sizes were unchanged compared to IDR$_{WT}$, demonstrating that cation-pi and hydrophobic interactions are not required for condensate formation (S3 Fig). Together, these results support the N-terminal IDR of p26 inducing phase separation through electro-static interactions.

## Charged residues govern p26 nucleolar partitioning

Umbravirus movement proteins must access the nucleolus to support systemic virus trafficking [45]. Nucleolar partitioning of full-length p26$_{WT}$, p26$_{R/K-G}$, p26$_{ΔNLS}$, or p26$_{D/E-G}$ was examined by expressing proteins from a CaMV 35S promoter in *N. benthamiana* (via agroin-filtration) and observing localization in cells with DAPI-stained nuclei. As with ORF3 ortholo-gues [40, 44, 45, 57], p26$_{WT}$ was observed in nuclear bodies (e.g., the nucleolus) in addition to forming cytoplasmic granules (Fig 3A). The percentage of nuclear granules was calculated by manually counting nuclear granules from six 20x fields and dividing by the total granule count calculated using the ImageJ "analyze particles" function in the 488 nm channel. Approximately 5% of p26$_{WT}$ granules were nuclear and coincided with published observations for the related GRV pORF3 [57]. p26$_{R/K-G}$ did not form granules but instead was diffusely expressed through-out the cytoplasm and failed to partition in the nucleolus (Fig 3A). p26$_{ΔNLS}$ resulted in strictly cytoplasmic localization of p26 granules (Fig 3A and 3B). p26$_{D/E-G}$ formed cytoplasmic gran-ules similar to those of p26$_{WT}$ (Fig 3A), despite reduced phase separation of IDR$_{D/E-G}$ *in vitro*. However, the portion of p26$_{D/E-G}$ granules that were nuclear (33%) was 6.5-fold higher com-pared to p26$_{WT}$ (Fig 3B), suggesting that the net charge of p26 may be influencing nucleolar localization.

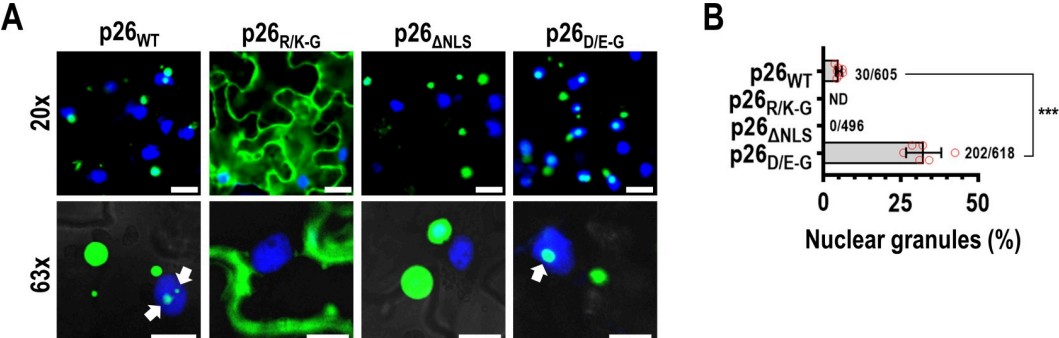

**Fig 3. Charged residues govern p26 nucleolar partitioning.** (A) p26-GFP fusions were expressed from the CaMV 35S promoter in *N. benthamiana* leaves following agroinfiltration. Prior to imaging, leaves were infiltrated with 5 μg/mL DAPI to stain nuclei. 20x and 63x fields are shown. Arrows denote p26 partitioned inside nuclear bodies (e.g. nucleolus). Bar scale: Top 20 μm; Bottom 10 μm. Images in all panels are representative of two independent experiments. (B) Nuclear granules were manually counted from six 20x fields across three biological replicates. Total granule counts (>2 μm$^2$ in size) were counted using the ImageJ "analyze particles" tool. Error bars denote standard deviations and data points (red circles) denote individual 20x fields. ****$P$<0.0001 unpaired t test, ND not detected.

## p26 phase separation is required for partitioning into Fib2 droplets

Fibrillarin (Fib2), which makes up the dense fibrillar component of the nucleolus [58], is required for systemic trafficking of umbravirus vRNPs [43, 44]. The N-terminus of *Arabidopsis thaliana* Fib2 (amino acids 7–77) comprises an intrinsically disordered glycine- and arginine-rich (GAR) domain (Fib2$_{GAR}$) (Fig 4A) that is common to fibrillarin across eukaryotes [59]. To determine whether Fib2$_{GAR}$ is sufficient for Fib2 phase separation, histidine-tagged full-length Fib2 (Fib2$_{FL}$) or Fib2$_{GAR}$ were fused to the N-terminus of mCherry and purified from *E. coli* for *in vitro* phase separation assays (Fig 4B). mCherry alone did not form droplets in the presence of 10% PEG-8000 or under high-salt conditions (Fig 4C and 4D), whereas Fib2$_{FL}$ and Fib2$_{GAR}$ readily formed droplets under crowding conditions (Fig 4C). In the presence of 1 M NaCl, Fib2$_{FL}$ droplets, but not those of Fib2$_{GAR}$ were resistant to high-salt (Fig 4C and 4D). These results indicate that the GAR domain is sufficient to form Fib2 droplets through electrostatic interactions and is consistent with findings using mammalian and *Caenorhabditis elegans* fibrillarin [49, 60, 61]. In addition, unlike Fib2$_{GAR}$, Fib2$_{FL}$ condensates are either not strictly dependent on electrostatic interactions or Fib2$_{FL}$ can form salt-resistant aggregates.

Fib2 functions as a scaffold for recruiting client proteins into the nucleolus and scaffolds are typically present in excess relative to clients [62, 63]. During viral infection, p26 is thought to partition into already formed nucleolar Fib2 droplets [49] and the GRV p26 ortholog can

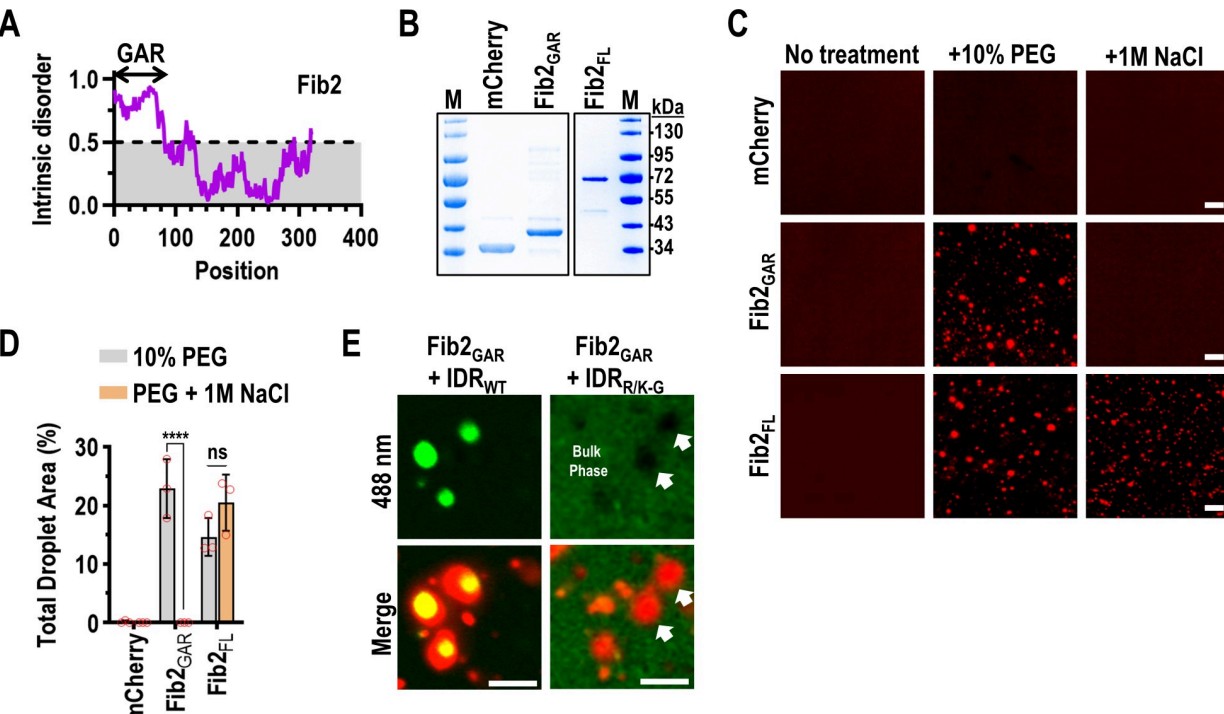

**Fig 4. p26 phase separation is required for partitioning into Fib2 droplets.** (A) Graphical representation of predicted intrinsic disorder in *A. thaliana* Fib2 using IUPRED [53]. The N-terminal glycine and arginine rich (GAR) domain is labelled. (B) The Fib2 GAR domain (Fib2$_{GAR}$) and full-length Fib2 (Fib2$_{FL}$) were fused to mCherry and purified from *E. coli* and analyzed by SDS-PAGE. Proteins were Coomassie stained and molecular weight (kDa) marker is shown. (C) mCherry, Fib2$_{GAR}$, and Fib2$_{FL}$ were examined by confocal microscopy after inducing phase separation with 10% PEG-8000 alone or in the presence of 1 M NaCl. Eight micromolar protein was used for all assays. Bar scale: 20 μm. Experiments were repeated. (D) Total droplet areas of Fib2$_{GAR}$ and Fib2$_{FL}$ were measured using ImageJ. Error bars denote standard deviations and data points (red circles) represent representative 20x fields (3 total) for each condition. **** *P*<0.0001, ns: not significant using two-way ANOVA with Sidak's multiple comparisons test. (E) Fib2$_{GAR}$ droplets were pre-formed using 24 μM protein before the addition of 4 μM IDR$_{WT}$ or IDR$_{R/K-G}$. Sorting of IDR$_{WT}$ to Fib2 droplets was observed by confocal microscopy. White arrows indicate exclusion of IDR$_{R/K-G}$ from pre-formed Fib2$_{GAR}$ droplets. Bar scale 10 μm. Images in all panels are representative of two independent experiments.

directly interact with $Fib2_{GAR}$ [44]. Using a 1:6 molar ratio of $IDR_{WT}$ and $Fib2_{GAR}$, $IDR_{WT}$ readily partitioned into pre-formed $Fib2_{GAR}$ droplets *in vitro* (Fig 4E, Left). When $IDR_{R/K-G}$ was added to pre-formed $Fib2_{GAR}$ droplets, $IDR_{R/K-G}$ remained in the bulk phase and was excluded from $Fib2_{GAR}$ droplets (Fig 4E, Right, white arrows). These results demonstrate that p26 phase separation is critical for partitioning into Fib2 droplets, which could play a role in PEMV2 movement.

## vRNPs required for systemic trafficking can be reconstituted via phase separation

Movement-competent umbravirus vRNPs consist of Fib2, p26, and the genomic RNA (gRNA) [44]. Therefore, we sought to determine whether vRNPs could be re-constituted *in vitro* through phase separation. To determine whether full-length PEMV2 gRNA was sorted to Fib2 droplets, Cy5-labelled PEMV2 gRNA was mixed with pre-formed $Fib2_{GAR}$ or $Fib2_{FL}$ droplets at a 1:500 RNA:Fib2 molar ratio (this ratio was used since earlier work determined that umbravirus RNAs were saturated by protein interactors under these conditions [41, 44]). PEMV2-Cy5 gRNA was not efficiently sorted into $Fib2_{GAR}$ droplets when visualized by confocal microscopy (Fig 5A) or measured using Mander's Overlap Coefficient (MOC, Fig 5B), which is consistent with earlier findings that determined $Fib2_{GAR}$ does not bind RNA [59, 60]. In contrast, $Fib2_{FL}$ captured a significantly higher portion of PEMV2-Cy5 gRNA demonstrating that PEMV2 gRNA can efficiently partition into full-length Fib2 droplets (Fig 5A and 5B). Since p26 must also associate with viral RNAs, PEMV2-Cy5 gRNA was mixed with pre-formed $IDR_{WT}$ droplets using a 1:500 ratio of RNA to protein. Approximately 50% of the $IDR_{WT}$ signal spatially overlapped the PEMV2-Cy5 signal when visualized by confocal microscopy and quantified by MOC (Fig 5C and 5D). Partitioning of PEMV2 gRNA into $IDR_{WT}$ droplets was not unique to PEMV2 since the distantly related carmovirus *Turnip crinkle virus* (TCV) and non-viral *Renilla* luciferase (RLuc) RNAs also sorted into $IDR_{WT}$ droplets with equal propensity (Fig 5C and 5D). Importantly, the N-terminal His-tag of $IDR_{WT}$ did not influence sorting of RNAs into droplets (S1 Fig). Finally, equimolar amounts of $Fib2_{FL}$ and $IDR_{WT}$ (8 μM each) were mixed with PEG, followed by the addition of 16 nM PEMV2-Cy5 gRNA, which resulted in the formation of droplets containing $IDR_{WT}$, $Fib2_{FL}$, and PEMV2 gRNA (Fig 5E). Together, these findings suggest that the formation of vRNPs *in planta* may occur in a dense phase with resident p26, Fib2, and viral RNAs.

## Phase separation-deficient p26 mutants fail to systemically traffic a virus vector

Since p26 protects PEMV2 subgenomic RNA from nonsense-mediated decay [42], it is required for efficient PEMV2 RNA replication in single cells [64]. Thus, it is difficult to distinguish the effects of p26 mutations on RNA replication and virus movement. To separate these two functions, we assayed if p26 mutants could support movement of a heterologous viral RNA in *N. benthamiana*. The TMV TRBO vector contains a coat protein (CP) deletion that allows cell-to-cell movement but not systemic movement [65], which can be rescued by umbravirus ORF3 proteins expressed *in trans* or from a subgenomic promoter [39, 66]. p26 can also systemically move TRBO when expressed as a GFP-fusion, which allows for visual monitoring of virus movement [42].

Local infections were established following agroinfiltration of *N. benthamiana* plants (4th leaf stage) using TRBO vectors expressing either free GFP, $p26_{WT}$, $p26_{R/K-G}$, or $p26_{D/E-G}$ (Fig 6A). High levels of free GFP and lower levels of wild-type or mutant p26 were observed at 4 days post-infiltration (dpi) by visual fluorescence and western blotting using anti-GFP

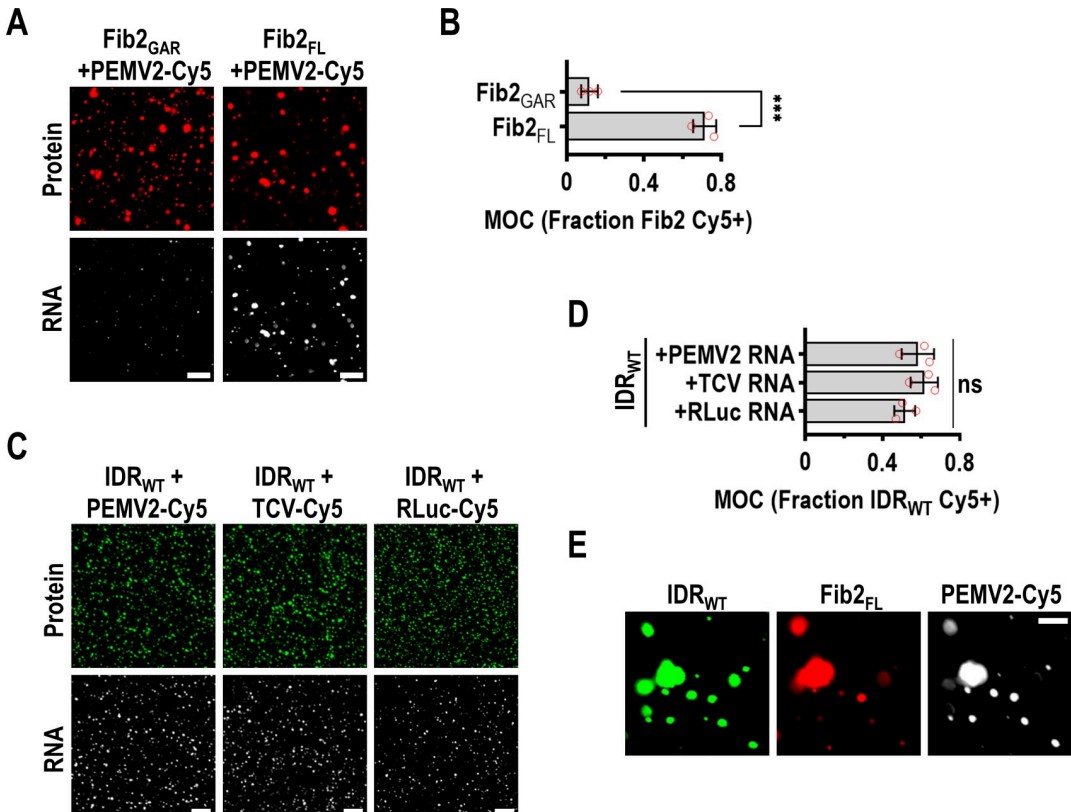

**Fig 5. vRNPs required for systemic trafficking can be reconstituted *in vitro* via phase separation.** (A) Fib2$_{GAR}$ and Fib2$_{FL}$ droplets were pre-formed prior to the addition of PEMV2-Cy5 gRNAs at a 1:500 RNA:protein molar ratio. Sorting of Cy5-labelled RNAs into Fib2 droplets was monitored using confocal microscopy. Bar scale: 20 μm. (B) The fraction of Fib2$_{GAR}$ or Fib2$_{FL}$ signal that was positive for Cy5-labelled RNA was determined by MOC analysis using EzColocalization [87]. Error bars denote standard deviations and individual data points (red circles) represent individual 20x fields (3 total) for each condition. ***$P<0.001$ unpaired t test. (C) IDR$_{WT}$ droplets were pre-formed prior to the addition of PEMV2-Cy5 gRNA, TCV-Cy5 gRNA, or RLuc-Cy5 RNA. Bar scale: 20 μm. (D) The fraction of IDR$_{WT}$ signal that was positive for Cy5-labelled RNA was determined by MOC analysis. ns: not significant by unpaired t test. Error bars denote standard deviations. Three 20x fields were quantified for each condition (red circles). (E) IDR$_{WT}$, Fib2$_{FL}$, and PEMV2-Cy5 gRNA were mixed under crowding conditions. Bar scale: 10 μm. Images in all Fig 5 panels are representative of at least two independent experiments.

antibodies (Fig 6B). Localization patterns of p26$_{WT}$ and p26$_{D/E-G}$ were unchanged when expressed from either a 35S promoter or from the TRBO vector and supported the previous finding that p26$_{D/E-G}$ granules were significantly enriched in nuclei compared to p26$_{WT}$ during virus infection (Fig 6C). Systemic movement of TRBO expressing p26$_{WT}$ was visually apparent by 14 dpi and was confirmed by RT-PCR whereas TRBO did not move systemically when expressing free GFP (Fig 6D). p26$_{R/K-G}$, which did not phase separate or enter the nucleolus, was unable to support TRBO movement (Fig 6D). Surprisingly, p26$_{D/E-G}$ also failed to support TMV movement despite the ability of IDR$_{D/E-G}$ to form droplets (albeit less efficiently than IDR$_{WT}$) and partition into the nucleolus. One possibility is that increased nucleolar retention of p26$_{D/E-G}$ contributed to the block in systemic movement, which would imply that nucleolar and virus trafficking by p26 is a tightly regulated process. Together, these data suggest that p26 phase separation, nucleolar partitioning, and virus movement are connected and co-dependent on charged residues. Although deletion of the TMV CP was previously reported to block systemic movement of the TRBO vector [65], we routinely observed systemic trafficking of TRBO-GFP after 3 weeks (S4 Fig). However, TRBO-GFP was restricted to the petiole and

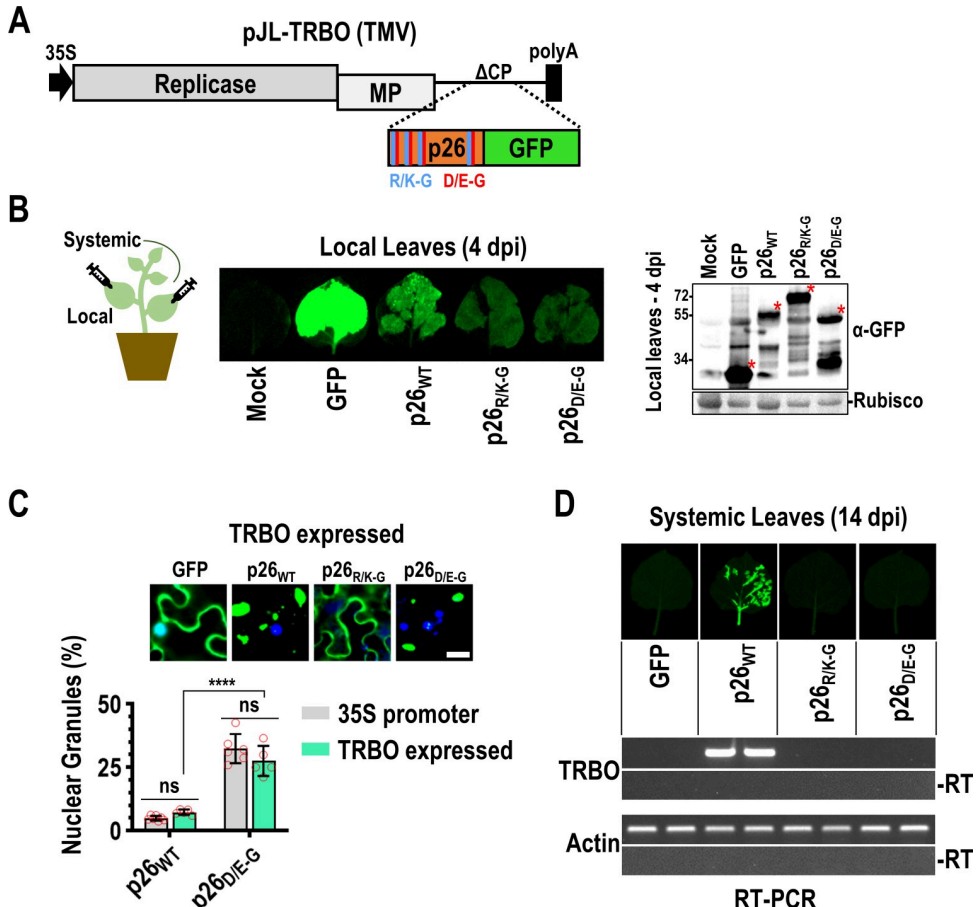

**Fig 6. Phase separation-deficient p26 mutants fail to systemically traffic a virus vector.** (A) The TMV-derived TRBO vector lacks CP and is severely impaired in systemic trafficking. Free GFP, p26$_{WT}$, p26$_{R/K-G}$, and p26$_{D/E-G}$ GFP-fusions were expressed from TRBO after establishing local infections via agroinfiltration (B) GFP-fusion proteins were visualized and detected in local leaves at 4 dpi by UV exposure (Left) or western blotting (Right). Pooled samples from 3 biological replicates were used for western blotting. Rubisco serves as a loading control. Red asterisks denote free GFP or GFP-fusion bands. (C) Localization patterns of free GFP, p26$_{WT}$, p26$_{R/K-G}$, and p26$_{D/E-G}$ in TRBO-infected leaves. Nuclear p26$_{WT}$ or p26$_{D/E-G}$ granules were counted from five 20x fields across three biological replicates and divided by the total number of granules (counted with ImageJ) to calculate a percentage (%). Bar scale: 20 μm. Results were compared with p26$_{WT}$ or p26$_{D/E-G}$ expressed from the duplicated 35S promoter (Fig 3B data is included for comparison). Error bars denote standard deviations and data points (red circles) represent individual 20x fields. ns not significant by multiple unpaired t tests. ****$P < 0.0001$ unpaired t test. (D) Systemic leaves were imaged at 14 dpi. RT-PCR was used to detect the TRBO vector or actin as a control. -RT: No reverse transcriptase controls. Two pools of 3–4 leaves are shown for each construct. Results are representative of three independent experiments consisting of at least 4 plants/construct.

midrib of systemic leaves whereas TRBO-p26$_{WT}$ spread throughout the veins and invaded the lamina (S4 Fig).

## p26 sorts into G3BP phase separations that restrict PEMV2 accumulation

Since SGs can have both proviral and antiviral roles in RNA virus infection cycles, we investigated whether p26 could partition into G3BP SGs. A nuclear transport factor 2 (NTF2) protein with RNA recognition motif (RRM) from *A. thaliana* functions as a G3BP-like SG nucleator in plants [67] (Fig 7A). As previously demonstrated by Krapp et. al. [67], RFP-tagged G3BP displays a diffuse cytoplasmic expression pattern in *N. benthamiana* under no stress but forms

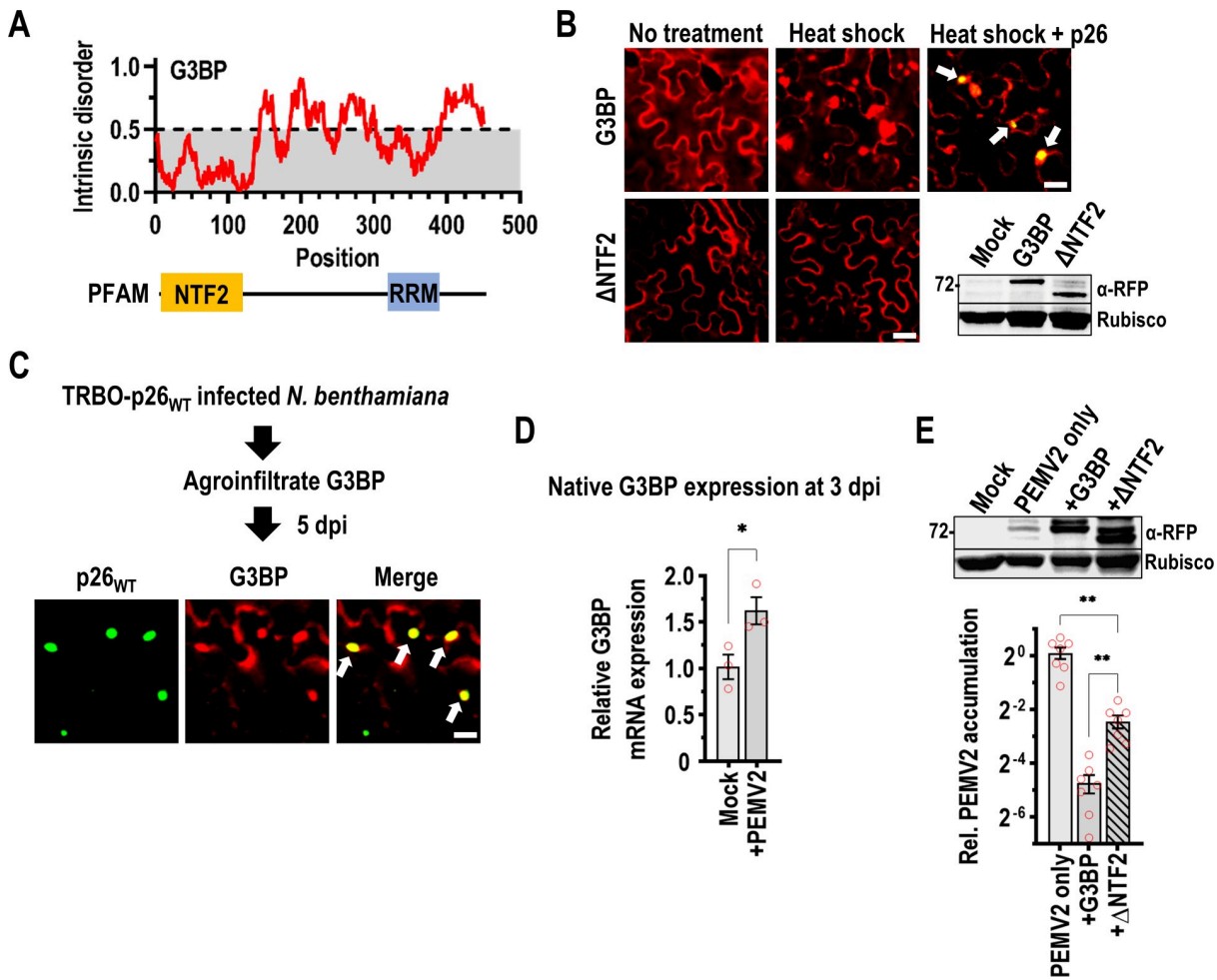

**Fig 7. p26 is sorted into G3BP phase separations that restrict PEMV2 accumulation.** (A) Graphical representation of predicted intrinsic disorder in *A. thaliana* G3BP using IUPRED [53]. G3BP contains an N-terminal NTF2 domain and C-terminal RNA Recognition Motif (RRM). (B) Following agroinfiltration, G3BP or ΔNTF2 expression patterns were visualized at 3 dpi in the absence of stress or after heat shock. During co-expression, p26 partitioning into G3BP SGs was observed following heat shock (White arrows). Scale bar: 20 μm. Inset shows western blot using anti-RFP antibodies to detect full-length G3BP and ΔNTF2. Rubisco was used as a loading control. Results represent two independent experiments. (C) G3BP was agroinfiltrated into *N. benthamiana* plants systemically infected with TRBO-p26$_{WT}$. Confocal microscopy was used to observe co-localization (White arrows) between p26 and G3BP during virus infection. Scale bar: 20 μm. Results are representative of two independent experiments. (D) Native G3BP expression was measured in Mock- or PEMV2-infected *N. benthamiana* at 3 dpi by RT-qPCR. The agroinfiltrated p14 RNA silencing suppressor was used as a reference gene. Data is from three biological replicates (red circles). *$P < 0.05$; student's t-test. Bars denote standard error. (E) PEMV2 was agroinfiltrated alone, or alongside either G3BP or ΔNTF2 (both tagged with RFP). Western blot confirmed expression of G3BP and ΔNTF2 (top). RT-qPCR was used to measure PEMV2 gRNA accumulation and represents 7 biological replicates from 2 independent experiments (red circles, Bottom). Bars denote standard error. Brown-Forsythe and Welch ANOVA with multiple comparisons was used to determine if observed differences were significant. ** $P < 0.01$.

cytoplasmic SGs after heat shock (Fig 7B). In mammals, the N-terminal NTF2 domain is required for both G3BP phase separation and recruitment to SGs [68, 69] and here ΔNTF2 failed to form SGs after heat shock (Fig 7B). During co-expression of p26$_{WT}$ and G3BP from 35S promoters, p26 partitioned into SGs following heat shock as visualized by confocal microscopy (Fig 7B, white arrows).

To determine whether p26 partitions into SGs during a viral infection, G3BP was agroinfiltrated into *N. benthamiana* plants systemically infected with TRBO-p26$_{WT}$ (Fig 7C). p26 condensates co-localize with G3BP demonstrating that p26 and G3BP can share phase separations

during an authentic viral infection (Fig 7C). When native G3BP expression was measured by RT-qPCR at 3 dpi in PEMV2-infected *N. benthamiana* leaves, a 61% increase in expression was observed during infection that could be part of an antiviral host response (Fig 7D). To confirm that G3BP has an inhibitory effect on PEMV2 accumulation, G3BP was co-infiltrated with PEMV2 and the p14 silencing suppressor into *N. benthamiana*. At 3 dpi, PEMV2 accumulation was reduced >20-fold by G3BP over-expression, demonstrating that G3BP exerts strong antiviral activity (Fig 7E). Virus accumulation was partially restored (only 5-fold inhibition) during overexpression of ΔNTF2 indicating that phase separation by G3BP is required for maximal antiviral activity (Fig 7E).

## Discussion

Our study demonstrates that p26 forms viscous droplets *in vivo* that fail to recover more than 50% in FRAP assays suggesting that intra-droplet dynamics are poor. Using *in vitro* assays, we determined that the N-terminal IDR of p26 drives phase separation through electrostatic interactions since droplet formation was significantly reduced in the presence of high-salt. Furthermore, mutation of all basic residues to glycine (R/K-G), prevented p26 droplet formation both *in vitro* and *in vivo*. $p26_{R/K-G}$ was also unable to systemically move a TMV vector lacking CP but it remains unclear whether the block in systemic movement was due to the inability of $p26_{R/K-G}$ to phase separate or to enter the nucleolus, or a combination of both. Since p26 must interact with Fib2 in phase-separated nucleoli to support virus movement [44], we also investigated whether $IDR_{WT}$ or $IDR_{R/K-G}$ could partition into pre-formed Fib2 droplets. Unlike $IDR_{WT}$, $IDR_{R/K-G}$ remained in the bulk-phase and was excluded from Fib2 droplets suggesting that the ability of p26 to assemble in a dense phase is important for interacting with Fib2.

While mutation of acidic residues (D/E-G) did not abolish phase separation *in vitro*, phase separation was significantly reduced compared to wild-type as measured by turbidity, total droplet area, and mean droplet size. Phase separation of arginine-rich peptides can occur through charge repulsion in the presence of buffer counteranions or RNAs [70, 71], which could explain how arginine-rich $p26_{D/E-G}$ forms droplets. Nucleolar retention of $p26_{D/E-G}$ granules was 6.5-fold higher compared to $p26_{WT}$, which might have resulted from increased protein net charge ($p26_{D/E-G}$ at pH 7.4 is +36 compared to +14 for $p26_{WT}$). These findings support earlier work showing that nucleolar localization of cellular and viral proteins was dependent on the overall positive charge [72, 73]. Surprisingly, $p26_{D/E-G}$ failed to support movement of a TMV vector, which may have resulted from increased nucleolar retention.

SGs that are assembled by self-association and phase separation of G3BP can support or restrict RNA virus replication [68, 69]. Seven *A. thaliana* G3BP-like candidates have been identified [74] containing an N-terminal NTF2 domain that is required for phase separation of mammalian G3BP1 [69]. G3BP expression was upregulated during PEMV2 infection suggesting that G3BP could be expressed as part of a concerted host response to infection. We determined that PEMV2 RNA accumulation was severely restricted by over-expression of G3BP but was partially restored during expression of ΔNTF2, demonstrating that phase separation of G3BP enhances antiviral activity towards PEMV2.

Since PEMV2 accumulation was not fully restored during ΔNTF2 expression, G3BP retains measurable antiviral activity in the dilute state in the absence of SGs. Human G3BP1 can bind and promote the degradation of mRNAs with structured 3' untranslated regions (3' UTRs) in conjunction with upframeshift 1 (Upf1) as part of the structure-mediated RNA decay (SRD) pathway [75]. PEMV2 contains a highly structured 3' UTR [76] and like many RNA viruses is inhibited by Upf1 [77, 78]. Therefore, G3BP over-expression could enhance SRD targeting of PEMV2 RNAs. While it remains unknown whether p26 partitioning into G3BP SGs is

beneficial or detrimental for PEMV2 replication, p26 disrupts the Upf1-dependent nonsense-mediated decay (NMD) pathway [42] and Upf1 is known to partition into SGs [79]. Future research will investigate whether partitioning of p26 into SGs interferes with Upf1- or G3BP-dependent RNA decay pathways.

In summary, our findings demonstrate that a plant virus movement protein forms droplets and partitions inside the nucleolus and SG membraneless compartments. Since nucleolar partitioning is required for virus trafficking and G3BP SG formation severely restricts PEMV2 replication, our findings highlight both beneficial and detrimental virus-host interactions mediated by phase separation.

## Materials & methods

### Construction of plant and bacterial expression vectors

All coding sequences and cloning strategies for the following constructs are detailed in S1 Appendix. Sequences for all primers used in this study are available in S1 Table. The pBIN61S binary vector was used to express proteins of interest in plants from the constitutive *Cauliflower mosaic virus* (CaMV) duplicated 35S promoter. $p26_{WT}$, $p26_{R/K-G}$, $p26_{D/E-G}$, and $p26_{\Delta NLS}$ GFP-fusions were PCR-amplified from synthetic double-stranded DNA fragments (Integrated DNA Technologies, Coralville, Iowa) and cloned into pBIN61S using the *BamH*I and *Sal*I restriction sites. $p26_{R/K-G}$ and $p26_{D/E-G}$ constructs contain glycine substitutions for all basic or acidic p26 residues, respectively. $p26_{\Delta NLS}$ is missing the coding sequence for amino acids 100–105 (5'-RRRARR-3') of p26. pBIN61S-GFP has been previously described [80]. The TMV vector pJL-TRBO has been previously described [65] and was a gift from John Lindbo (Addgene plasmid # 80082). pJL-TRBO containing $p26_{WT}$ fused to GFP has also been previously described [42]. $p26_{R/K-G}$ and $p26_{D/E-G}$ GFP-fusions were PCR amplified from synthetic DNA fragments with introduced *Pac*I and *Not*I restriction sites for digestion and ligation into the corresponding pJL-TRBO sites. G3BP:RFP contains the G3BP coding region (accession AT5G43960.1) followed by the coding region of RFP and was a generous gift from Dr. Björn Krenz [67]. To construct ΔNTF2, G3BP:RFP was PCR amplified with the coding sequence for amino acids 2–125 of G3BP omitted. PCR amplification introduced forward *BamH*I and reverse *Sal*I restriction sites for cloning into pBIN61S. All ligations used T4 DNA Ligase (New England Biolabs, Ipswich, Massachusetts) following the manufacturer's protocol. pCB-PEMV2 has been previously described [77] and contains the full-length PEMV2 genome under control of a CaMV duplicated 35S promoter and was used to establish infections in *N. benthamiana*. All constructs were Sanger sequenced for accuracy.

For C-terminal GFP-fusion recombinant protein production in *E. coli*, pRSET his-eGFP [81] was used as a backbone and was a gift from Jeanne Stachowiak (Addgene plasmid # 113551). Coding sequences for the wild-type p26 IDR (amino acids 1–132) or p26 C-terminus (amino acids 133–226) were PCR amplified from a full-length PEMV2 infectious clone. The coding sequence for the last 10 amino acids of p26 was omitted from the C-term construct to circumvent proteolysis encountered during bacterial expression. Mutant IDRs containing R/K-G, D/E-G, or ΔNLS mutations were synthesized (Integrated DNA Technologies, Coralville, Iowa) as double-stranded DNA fragments and were used in restriction digests and ligation reactions. $IDR_{WT}$ was cloned into the *BamH*I restriction site of pRSET his-eGFP and sequenced for directionality and accuracy. C-term, $IDR_{R/K-G}$, $IDR_{D/E-G}$, and $IDR_{\Delta NLS}$ were cloned into pRSET his-eGFP using both the *Nhe*I and *BamH*I restriction sites and sequenced for accuracy.

Fib2 (accession AT4G25630.1) was cloned by first synthesizing cDNAs from *A. thaliana* seedling total RNAs using random hexamers and SuperScript III reverse transcriptase

according to the manufacturer's protocol (ThermoFisher Scientific, Waltham, Massachusetts). Next, the Fib2 coding sequence was PCR amplified using primers that introduced *Nhe*I and *Bam*HI restriction sites for cloning the fragment into pRSET-his-mCherry [82], a gift from Jeanne Stachowiak (Addgene plasmid # 113552). The resulting construct was full-length Fib2 with a C-terminal mCherry fusion (Fib2$_{FL}$). The coding sequence for the Fib2 GAR domain (amino acids 7–77 of Fib2) was PCR amplified from Fib2$_{FL}$ using primers that introduced *Nhe*I and *Bam*HI restriction sites, digested, and ligated into corresponding sites of pRSET-his-mCherry to generate Fib2$_{GAR}$. Both constructs contain N-terminal histidine tags for affinity purification and were sequenced for accuracy.

## Agroinfiltration and plant growth

All plant expression constructs used in this study were electroporated into *Agrobacterium tumerfaciens* strain C58C1. Liquid cultures were passaged in media containing the appropriate antibiotics and 20 μM acetosyringone 1 day prior to infiltration. Overnight cultures were pelleted and resuspended in a solution of 10 mM $MgCl_2$, 10 mM MES-K [pH 5.6], and 100 μM acetosyringone as previously described [77]. All agroinfiltrations included a construct expressing the p14 RNA silencing suppressor from *Pothos latent virus* and has been previously described [50]. p14 was agroinfiltrated using a final $OD_{600}$ of 0.2 and was included to enhance transient gene expression. pBIN61S-derived constructs expressing GFP, p26$_{WT}$, p26$_{R/K-G}$, p26$_{\Delta NLS}$, or p26$_{D/E-G}$ were agroinfiltrated at a final $OD_{600}$ of 0.4. pJL-TRBO TMV vectors expressing GFP, p26$_{WT}$, p26$_{R/K-G}$, or p26$_{D/E-G}$ were also agroinfiltrated at a final $OD_{600}$ of 0.4. G3BP and ΔNTF2 were agroinfiltrated using a final $OD_{600}$ of 0.2. Finally, pCB-PEMV2 was agroinfiltrated alone or co-infiltrated with G3BP or ΔNTF2 using a final $OD_{600}$ of 0.2 for each construct. Typically, the 3$^{rd}$-5$^{th}$ leaves from young *N. benthamiana* plants were infiltrated with a 1 mL syringe. Visualization of nuclei in agroinfiltrated leaves was achieved by infiltrating a solution of 5 μg/mL DAPI (4′,6-diamidino-2-phenylindole) into leaves 45 minutes prior to confocal imaging. *N. benthamiana* plants were grown in a humidity-controlled chamber at 24°C, 65% humidity, and 12-hour day/night schedule (200 μmol m$^{-2}$s$^{-1}$).

## Fluorescence recovery after photobleaching (FRAP)

p26$_{WT}$ was transiently expressed from a 35S promoter in *N. benthamiana* as a GFP-fusion and produced visible fluorescence by 2 dpi. Leaf samples were wet-mounted and imaged using a Zeiss LSM 510 Meta confocal microscope with a 20X objective. FRAP was performed using Zen 2009 software (Zeiss, Oberkochen, Germany) and photobleaching a ~2 μm diameter region with 100% laser power (488 nm) with subsequent fluorescence recovery measured at 5 s intervals. Background regions and unbleached reference droplets were recorded as controls. Data analysis was performed by subtracting background intensities, followed by normalization to set the first post-bleach value to zero as previously described [83]. Seven p26$_{WT}$ droplets were analyzed by FRAP and data was presented as a fraction of the pre-bleach fluorescence intensity with standard deviations using GraphPad Prism software (GraphPad, San Diego, California).

## Protein expression and purification

The protein sequences of all recombinant proteins produced in this study are available in S1 Appendix. Histidine-tagged recombinant proteins were expressed in BL21(DE3) *E. coli* (New England Biolabs, Ipswich, Massachusetts) using autoinduction Luria-Bertani (LB) broth and purified using HisPur cobalt spin columns (Thermo Scientific, Waltham, Massachusetts). Proteins were purified under denaturing conditions according to the manufacturer's protocol

using 8 M urea. All equilibration, wash, and elution buffers contained 1 M NaCl to suppress phase separation. Following elution of recombinant proteins from the cobalt resin, proteins were re-folded through dialysis in buffer containing 10 mM Tris-HCl (pH 7.0), 300 mM NaCl, 1 mM EDTA, 1 mM dithiothreitol, and 10% glycerol as previously used for the related pORF3 from GRV [41]. Urea was removed in a stepwise fashion by using dialysis buffers containing 4 M urea, 1 M urea, or no urea. Proteins were concentrated using Amicon 10K Ultra centrifugal filters and concentrations were measured using a Bicinchoninic acid (BCA) protein assay (Millipore Sigma, St. Louis, Missouri). Next, protein molarity was determined using predicted molecular weights for each protein using ProteinCalculator v3.4 (http://protcalc.sourceforge.net/) and are available in S1 Appendix. Protein purity was assessed by SDS-PAGE. If necessary, hydrophobic interaction chromatography using methyl HIC resin was used to further purify and concentrate GFP-fusion samples according to the manufacturers protocol (Bio-Rad, Hercules, California).

### In vitro phase separation and RNA sorting assays

For *in vitro* assays, recombinant proteins were used at a final concentration of 8 μM unless otherwise noted. Phase separation assays consisted of the following mixture: 8 μM protein, 10 mM Tris-HCl (pH 7.5), 1 mM DTT, 100 mM NaCl, and 10% PEG-8000 to induce phase separation. High-salt conditions included NaCl at a final concentration of 1 M and "no treatment" did not include PEG-8000. Turbidity assays comparing $IDR_{WT}$ with controls or IDR mutants were performed with either 8 μM or 24 μM protein under standard assay conditions. 100 μL reactions were placed at room temperature for 15 minutes prior to $OD_{600}$ measurements using a 96-well plate reader. Details regarding *in vitro* assays using untagged $IDR_{WT}$ or tagged $IDR_{R-K}$ and $IDR_{VLIMFYW-S}$ constructs are available in S1 Materials and Methods.

Cy5-labelled gRNAs from PEMV2 or TCV were synthesized by T7 run-off transcription using *Sma*I-linearized full-length infectious clones of PEMV2 (see S1 Appendix) or TCV [84]. Cy5-labelled *Renilla* luciferase (RLuc) RNAs were synthesized from PCR products containing a T7 promoter, RLuc ORF, and a 13-nt 3' untranslated region using p2luci plasmid as template [85] (see S1 Table for primers). Cy5-UTP (APExBIO, Houston, Texas) was added to *in vitro* transcription reactions according to the HiScribe T7 Quick High Yield RNA Synthesis Kit protocol (New England Biolabs, Ipswich, Massachusetts). Synthesized RNAs were purified using the Monarch RNA Cleanup Kit (New England Biolabs, Ipswich, Massachusetts). RNA integrity was assessed by agarose gel electrophoresis and RNA concentrations were determined using a UV5Nano spectrophotometer. RNAs were mixed with 8 μM $IDR_{WT}$, $Fib2_{GAR}$, or $Fib2_{FL}$ at a final concentration of 16 nM (1:500 RNA:protein ratio). For observing vRNP formation *in vitro*, equimolar amounts of $Fib2_{FL}$ and $IDR_{WT}$ (8 μM each) were mixed with 16 nM PEMV2-Cy5 gRNA since atomic force microscopy revealed that Fib2 and GRV pORF3 form ring-like complexes with equimolar composition [43].

### Confocal microscopy and image processing

Phase separation during *in vitro* assays occurred rapidly and samples were directly loaded onto glass slides for confocal microscopy. A 20x objective from a Zeiss LSM 510 Meta confocal microscope with Zen 2009 software was used to visualize droplet formation. $IDR_{WT}$ and Fib2 droplets were observed after excitation with the 488-nm or 543-nm lines, respectively. Cy5-labelled RNAs were observed after excitation with the 633-nm laser line.

Confocal images were used to measure total droplet areas (%) from three representative 20x fields for each condition. Raw images (.lsm extension) were first imported into ImageJ [86]. Images were thresholded (typically 500–66535 for 16-bit images) and total droplet areas (%

Area) were measured using the "analyze particles" function excluding particles $<1$ $\mu m^2$ in size. Mean droplet sizes were measured from three representative 20x fields for $IDR_{WT}$, $IDR_{D/E-G}$, and $IDR_{\Delta NLS}$ using the "analyze particles" function. Particles $<0.2$ $\mu m^2$ in size were excluded from downstream analyses. Particle sizes were plotted using a cumulative distribution function (CDF) for comparing droplet sizes between $IDR_{WT}$ and mutant IDRs.

The number of p26 granules that co-localized with DAPI-stained nuclei after expression from a 35S promoter or TRBO vector in *N. benthamiana* were manually counted from 20x fields. Next, the total number of granules $>2$ $\mu m^2$ was counted for each thresholded 20x field using the ImageJ "analyze particles" function. Thresholds were used to include only granules in the focal plane of the 20x field. Dividing the number of nuclear granules by the total granule count yielded the percentage of nuclear granules for $p26_{WT}$, $p26_{\Delta NLS}$, or $p26_{D/E-G}$.

The ImageJ plugin EzColocalization [87] was used to measure colocalization in multi-channel confocal images. Raw images were imported into ImageJ and split into separate channels for GFP/mCherry and Cy5. Default settings including Costes threshold parameters were used to calculate Mander's overlap coefficients (MOC) that measured the fraction of $IDR_{WT}$, $Fib2_{GAR}$, $Fib2_{FL}$ signal that was positive for Cy5-labelled RNAs from three 20x fields.

## TMV movement assay and RT-PCR

Following agroinfiltration of TRBO vectors expressing GFP, $p26_{WT}$, $p26_{R/K-G}$, or $p26_{D/E-G}$ GFP-fusions, fluorescence in local and systemic leaves was monitored daily using a hand-held long-wave UV lamp. By 4 dpi, robust local infections were evident, and leaves were imaged (488 nm) using a Bio-Rad Gel Doc XR System prior to grinding in liquid nitrogen. Total protein was extracted by resuspending leaf tissue in 1X PBS supplemented with 3% β-mercaptoethanol and EDTA-free protease inhibitor cocktail (Thermo Scientific, Waltham, Massachusetts). Samples were mixed with 6X Laemmli SDS buffer, boiled, and separated by SDS-PAGE. A semi-dry transfer method was used to transfer proteins to nitrocellulose for western blotting using anti-GFP antibodies (Life technologies, Carlsbad, California) at a 1:5000 dilution. Goat anti-rabbit IgG (H+L) conjugated with horseradish peroxidase (Thermo Scientific, Waltham, Massachusetts) was used as a secondary antibody at a 1:5000 dilution. Blots were visualized using the Pierce enhanced chemiluminescence kit (Thermo Scientific, Waltham, Massachusetts). Systemic leaves were harvested at 14 dpi for total RNA extraction using Trizol. 100 ng total RNA was digested with RQ1 DNase (Promega, Madison, Wisconsin) and served as template for reverse transcription using iScript supermix (Bio-Rad, Hercules, California). No reverse transcriptase controls (-RT) were included for all sample and primer sets. 1 μL cDNA was used as template for 25 cycles of PCR using GoTaq polymerase (Promega, Madison, Wisconsin) targeting the TMV replicase in the TRBO vector. For loading controls, *N. benthamiana* actin was amplified by 31 cycles of PCR. Primer sequences are available in S1 Table.

## G3BP expression, SG formation, and visualization

Full-length G3BP or ΔNTF2 (RFP-tagged) were transiently expressed in *N. benthamiana* using agroinfiltration and visualized 3 dpi using confocal microscopy (543-nm line). Heat shock of G3BP-expressing plants was performed by incubating plants at 37°C for 45 minutes prior to imaging. p26 partitioning into SGs was observed following heat shock of plants transiently co-expressing $p26_{WT}$ and G3BP from 35S promoters. To observe co-localization of p26 and G3BP during virus infection, *N. benthamiana* plants (3–4 leaf stage) were first infiltrated with TRBO-$p26_{WT}$. After strong $p26_{WT}$ GFP-fusion signal was observed in the systemic leaves (typically ~2–3 weeks), G3BP was agroinfiltrated and leaves were imaged at 5 dpi following excitation with the 488-nm and 543-nm lines to observe p26 and G3BP, respectively. Using the same

protocol as above, western blotting with anti-RFP antibodies at a 1:5000 dilution (Thermo-Fisher Scientific, Waltham, Massachusetts) was performed to examine full-length G3BP or ΔNTF2 expression levels following agroinfiltration.

*RT-qPCR.* Agroinfiltrated "spots" were cut from leaves and stored at -80˚C. Samples were ground in liquid nitrogen and total RNA was extracted using the Quick-RNA Plant Kit (Zymo Research, Irvine, California). An on-column DNase I step was added using RQ1 DNase (Promega, Madison, Wisconsin). Total RNAs were used as templates for SYBR green-based one-step reverse-transcriptase quantitative PCR (RT-qPCR) using the NEB Luna One-Step RT-qPCR kit (New England Biolabs, Ipswich, Massachusetts). All primers used in this study were designed using Primer3 [88] and were validated by standard curve analysis with PCR efficiencies ranging from 90–110%. Targets included native *N. benthamiana* G3BP (Transcript ID: Niben101Scf03456g00002.1) and PEMV2 gRNA. Gene expression was normalized to the internal control transcripts from the agroinfiltrated p14 RNA silencing suppressor. All primer sequences are available in S1 Table. Expression analyses were performed by the ΔΔCq method using Bio-Rad CFX Maestro software. Target fidelity was monitored by melt curve analyses and no reverse transcriptase controls.

## Statistical analyses

GraphPad Prism (version 9.0.1) software was used for all statistical analyses in this study. Total droplet areas (%) from confocal images or turbidities ($OD_{600}$) were compared using a two-way ANOVA with Sidak's multiple comparison test. *P* values $<0.001$ were considered significant. Mann-Whitney tests were used to compare ranked droplet sizes ($\mu m^2$) from confocal images. *P* values $<0.001$ were considered significant. An unpaired t test was used to determine if differences in nuclear localization of $p26_{WT}$ or $p26_{D/E-G}$ granules were significant. Mander's Overlap Coefficient (MOC) values were compared using either an unpaired t test (2 samples, Fig 5B) or a one-way ANOVA with Tukey's multiple comparisons test (3 samples, Fig 5D). *P* values $>0.001$ were considered not significant. Nuclear localization of $p26_{WT}$ or $p26_{D/E-G}$ expressed from a 35S promoter or TRBO vector was compared using multiple unpaired t tests. *P* values $>0.001$ were considered not significant. Following RT-qPCR, relative G3BP gene expression values were compared using an unpaired t test. PEMV2 RNA levels were compared using Brown-Forsythe and Welch ANOVA tests and Dunnett's multiple comparisons test. *P* values $<0.05$ were considered significant.

## Supporting information

**S1 Fig. Characterization of His-tagged and untagged IDR$_{WT}$.** (A) Coomassie-stained SDS-PAGE gel shows expected subtle downward shift by IDR$_{WT}$ following His-tag cleavage with recombinant enterokinase (rEK). (B) Untagged or tagged IDR$_{WT}$ droplet formation was monitored under various conditions by confocal microscopy. Bar scale: 20 μm. (C) Total droplet areas (%) were measured from confocal images using ImageJ. Error bars denote standard deviations and data points represent individual 20x fields (3 total). ns: not significant by two-way ANOVA and Sidak's multiple comparisons test. (D) *In vitro* turbidity assays ($OD_{600}$) were performed with 8 μM tagged or untagged IDR$_{WT}$. Three biological replicates are shown (red circles). ns: not significant by unpaired t test. (E) Particle sizes of tagged and untagged IDR$_{WT}$ droplets from three 20x fields were measured using ImageJ. ns: not significant by two-tailed Mann-Whitney rank test. (F) Droplet dynamics of His-tagged and untagged IDR$_{WT}$ were measured by FRAP. Results are from 9 FRAP experiments with shaded areas representing standard deviations for each condition. Representative droplets and heat map overlays are shown for each construct. (G) End-point (2 min.) FRAP recoveries were compared between tagged and

untagged IDR$_{WT}$. Error bars denote standard deviations and data points represent individual FRAP experiments (9 total) [****]$P<0.0001$ unpaired t test. (H) RLuc-Cy5 RNAs were mixed with tagged and untagged IDR$_{WT}$ at a 1:500 RNA:protein ratio. The fraction of IDR$_{WT}$ signal that was positive for Cy5-labelled RNA was determined by Mander's Overlap Coefficient (MOC) analysis. Error bars denote standard deviations and data points represent individual 20x fields. ns: not significant by unpaired t test.
(TIF)

**S2 Fig. Aggregate formation by R/K-G.** (A) 24 μM protein was mixed with 10% PEG-8000 to induce phase separation in standard assay buffer. Droplet or aggregate formation was visualized by confocal microscopy. Bar scale: 5 μm. (B) Individual droplets or aggregates were assessed for circularity using ImageJ. Data points represent individual 20x fields and error bars denote standard deviations. [***]$P<0.001$, ns: not significant by one-way ANOVA with Dunnett's multiple comparisons test.
(TIF)

**S3 Fig. Cation-pi and hydrophobic interactions do not influence p26 phase separation.** (A) SDS-PAGE analysis of Coomassie-stained recombinant IDR$_{R-K}$ and IDR$_{VLIMFYW-S}$. Marker weights are shown on left in kilodaltons (kDa). IDR$_{R-K}$ contains lysine substitutions for all arginines whereas IDR$_{VLIMFYW-S}$ contains serine substitutions for all hydrophobic residues. (B) Droplet formation by IDR$_{WT}$, IDR$_{R-K}$, and IDR$_{VLIMFYW-S}$ was visualized by confocal microscopy under crowding conditions with and without 1 M NaCl. (C) Total droplet areas were measured from three separate 20x fields for each condition (red circles). Error bars denote standard deviations. ns: not significant by two-way ANOVA and Sidak's multiple comparisons test. (D) Turbidity assays (OD$_{600}$) comparing GFP, IDR$_{WT}$, IDR$_{R-K}$, and IDR$_{VLIMFYW-S}$ phase separation propensities. Error bars denote standard deviations and data points represent biological replicates (3 total). [****] $P<0.0001$ by two-way ANOVA with Dunnett's multiple comparisons test vs. IDR$_{WT}$. (E) Mean condensate sizes for IDR$_{R-K}$ and IDR$_{VLIMFYW-S}$ mutants and wild-type IDR$_{WT}$ were plotted by cumulative distribution frequency. Particle sizes were measured from three representative 20x fields using ImageJ. ns: not significant, two-tailed Mann-Whitney tests compared to IDR$_{WT}$.
(TIF)

**S4 Fig. Systemic trafficking of TRBO vector.** At 21 dpi, upper *N. benthamiana* systemic leaves were imaged at 488 nm. TRBO-GFP and TRBO-p26$_{D/E-G}$ were mostly restricted to the petiole and midrib of systemic leaves. In contrast, TRBO-p26$_{WT}$ invaded the lamina of systemic leaves. Images are representative of three independent experiments with at least four plants for each condition.
(TIF)

**S1 Appendix. Cloning details for all constructs generated in this study.** Details include recombinant protein amino acid sequences with predicted molecular weights, DNA coding sequences for all cloning inserts, and cloning strategies with restriction sites used. Full-length PEMV2 genome (not deposited in GenBank) is included.
(DOCX)

**S1 Table. Oligonucleotide sequences for primers used in this study.** Includes primers for cloning, *in vitro* transcription template preparation, RT-PCR, and RT-qPCR.
(DOCX)

**S1 Materials and Methods. Information on the construction of IDR$_{R-K}$ and IDR$_{VLIMFYW-S}$ along with purification details.** Details for Histidine-tag removal from IDR$_{WT}$, FRAP

analyses, image processing, and statistical analyses are included for data presented in supporting figures (S1–S4 Figs).
(DOCX)

## Acknowledgments

We would like to thank Dr. Björn Krenz (Leibniz Institut DSMZ, Brunswick, Germany) for the generous gift of the G3BP:RFP construct. We would also like to thank Dr. Jonathan Dinman and Dr. Anne Simon (University of Maryland) for their thoughtful insight. We would also like to thank Dr. Anne Simon for critical reading of the manuscript.

## Author Contributions

**Conceptualization:** Jared P. May.

**Formal analysis:** Jared P. May.

**Funding acquisition:** Jared P. May.

**Investigation:** Shelby L. Brown, Dana J. Garrison, Jared P. May.

**Methodology:** Shelby L. Brown, Jared P. May.

**Supervision:** Jared P. May.

**Validation:** Shelby L. Brown, Jared P. May.

**Visualization:** Jared P. May.

**Writing – original draft:** Jared P. May.

**Writing – review & editing:** Shelby L. Brown, Dana J. Garrison, Jared P. May.

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
