## [Decision Letter · Decision Letter 0]

31 May 2021

Dear Dr. May,

Thank you very much for submitting your manuscript "Phase separation of both a plant virus movement protein and cellular factors support virus-host interactions" for consideration at PLOS Pathogens. As with all papers reviewed by the journal, your manuscript was reviewed by members of the editorial board and by several independent reviewers. In light of the reviews (below this email), we would like to invite the resubmission of a significantly-revised version that takes into account the reviewers' comments.

We agree with all three reviewers that this is an interesting and important study. However, all reviewers identified substantial deficiencies in the current version, and provided constructive comments and suggested some additional experiments to improve the manuscript. Please follow these comments and suggestions to improve the manuscript. If you need more time for revision, please contact PLoS Pathogens editorial office. Your revised version will be reviewed by the editorial board and same reviewers.

We cannot make any decision about publication until we have seen the revised manuscript and your response to the reviewers' comments. Your revised manuscript is also likely to be sent to reviewers for further evaluation.

Sincerely,

Aiming Wang, Ph.D

Associate Editor

PLOS Pathogens

Shou-Wei Ding

Section Editor

PLOS Pathogens

Kasturi Haldar

Editor-in-Chief

PLOS Pathogens

orcid.org/0000-0001-5065-158X

Michael Malim

Editor-in-Chief

PLOS Pathogens

orcid.org/0000-0002-7699-2064

we agree with all three reviewers that this is an interesting and important study. However, all reviewers identified substantial deficiencies in the current version, and provided constructive comments and suggested some additional experiments to improve the manuscript. Please follow these comments and suggestions to improve the manuscript. If you need more time for revision, please contact PLoS Pathogens editorial office. Your revised version will be reviewed by the editorial board and same reviewers.

Reviewer's Responses to Questions

**Part I - Summary**

Reviewer #1: This paper introduces a really interesting new concept of phase separation playing a role in viral intercellular transport. The introduction explains that proteins containing intrinsically disordered regions self associate in oligomers, bind RNA, and phase separate when they are also enriched in arginine residues. The arginine residues are essential for cation-pi interactions with aromatic contacts to promote phase separation. Stress granules, the nucleolus represent examples of membraneless compartments in the cell and he suggests examples of cytoplasmic inclusions and some viral factores also aggregate as phase separation. This is a very interesting topic and this paper is the first to directly explore the concept for a plant virus, in this case PEMV2. The work is significant and novel and well executed overall. The in vitro turbidity assay and confocal microscopy are the strengths of the article, but there are some gaps when it comes to exlaining the various mutants in Figure2. Figure 2 is the most critical figure to the paper and there is room for improvement for this paper to be published. The M&M also needs to be better organized to match the order of the results, and the refererence list needs to be reviewed and edited for style.

Reviewer #2: In this manuscript, Brown and May present exciting data illustrating the phase separation of a viral protein and a host protein that participates in host-virus interactions. The viral protein, P26, participates in the phase separation in the nucleus with fibrillarin to support the systemic movement of viruses. P26 also phase separates together with a stress granule marker (G3BP) to limit viral accumulations when over-expressed. The study provides a timely update for the mechanistic understanding of host-virus interactions, thus fitting the scope of PLoS Pathogens.

Reviewer #3: This work presents a biochemical characterization of pea enation mosaic virus movement protein p26, which has an intrinsically disordered region with several charge amino acids at its N-terminal part. It belongs to proteins that can undergo phase separation both in vitro and in vivo. This property is convincingly demonstrated by many methods. With mutants having either all positively or all negatively charged amino acids of the N-terminal part substituted with glycine residues, the authors show that positive charges are required both for phase separation property and nuclear localization. Negative charges could be changed without affecting these functions, but the behavior of this protein in nucleus was altered. It’s association with nucleolus was prolonged which was presented as the possible reason for its failure to complement long-distance movement function of a movement-deficient TMV. The authors investigated the associations of p26 protein with fibrillarin and viral RNA and propose an interplay between these as an enabler of systemic movement. The participation of nucleolus and fibrillarin together with GRV, an umbravirus, movement protein has previously been studied in detail. The authors also predict an antiviral role for association between PEMV p26 and G3PB, which is manifested as a reduced PEMV accumulation upon G3BP upregulation. This subject should be studied further to demonstrate how G3BP actually interferes with PEMV 2 infection.

**Part II – Major Issues: Key Experiments Required for Acceptance**

Reviewer #1: Page 6 line 123—should explain that the in vitro assays start with gene expression in E. coli and explain what the assays are. The M&M does not have a subtitle for In vitro phase separation assays, so it is not explained there either. Also, figure 2C is a Coomassie gel, so I think the results may be an immunoprecipitation? The M&M suggest the constructs have His Tag and so does this impact the IDR assays because of their charged sidechains. This is very important to address. Notably PEG is used to precipitate proteins by absorbing water and so I would suggest that this is a turbidity assay, not necessarily functioning as a mimic of cell crowding. I suggest rephrasing lines 13-131 on page 6. Did free GFP also have the HisTag? Each lane in Fig 2C needs to be explained, for example what is R/K-G? I think the real in vitro assay is Figure 2D, not 2C. Importantly the authors indicate that IDR by itself is responsible for phase separation, but it would be useful to have additional segmental mutations to show that the non-IDR region is not responsible for phase separation. The R-K, VLIMFYW-S, R/K-G, and D/E-G are not defined in M&M or Figure 2 legend or results and these are central to testing the hypothesis.

Figure 2D is to show turbidity. An important control that is missing is the non-IDR region fused to GFP. The IDR-GFP fusion is not as green as GFP alone. Since I don’t know what R-K, VLIMFYW-S, R/K-G, and D/E-G, I am also wondering why these are not included in Figure 2D. What if you mixed other proteins or RNA into the in vitro system? Why not add the salt and PEG into the tubes in panel D as in panel E? Figure 2E is referred to as in vitro assay but it seems to be in vivo? OR is this solution placed on a slide?

The order of M&M sections should match the order of the results. The constructs start with the E coli expression vectors, but Figure 1 is Agro-infiltration of 35S plasmids. not E coli and I am not sure what the delivery is. It is not clear to me in Figure 1 and 2 if the p26 gene fusions are introduced into leaves via TMV vector or agro-delivery of plasmids. Lines 368 and 385 are contradictory—regarding synthesis and cloning. I suggest removing redundancies that may be confusing.

Pager 7 lines 144-146 describe Fig 2C which is out of order. Need to move the mutations up into the prior section and discuss all Figure 2 in one section of results.

Figure 3 is robust. But page 11 discusses figures out of order. I think this is confusing. Figure 4 shows P26-GFP complements movement defects of TMV which is a very important set of experiments to include.

Reviewer #2: Based on the presented data, P26 appears to facilitate viral systemic trafficking when phase separating with fibrillarin in the nucleus while phase separates with G3BP in cytoplasmic stress granule that seems to inhibit viral replication. But the data were all based on protein over-expressing. It will be informative to understand the P26 partition in the nuclear and cytoplasmic compartments in native infection conditions to quantitatively accessing the role of P26 in viral infection.

Reviewer #3: 1. Specificity of p26 functions in virus infection remains hard to interpret. Does its nucleolar and stress granule partitioning with fibrillarin and G3BP occur in a specific manner or is it typical for proteins with this kind of properties to co-aggregate at certain concentrations. Are there specific interactions of p26 with either Fib or G3BP? How is the selection of viral RNA done for long distance movement? Both cognate and non-cognate viral RNAs condensate with p26-GFP. Would any RNA condensate? The experimental design does not allow to make conclusions of how p26 works in PEMV infection.

2. The mutants used in this study are very robust. Changing all positively and all negatively charged amino acids to glycine alters the protein products drastically. The different functions p26 has in long distance movement (phase separation, nuclear localization, retention in the nucleus, protein-protein and protein-RNA interactions etc.) may become impossible to separate from each other.

I suggest that the specificity of p26 IDR region interactions be investigated by subtle mutations, and especially in the natural context of PEMV infection, to understand the requirements of PEMV long-distance transport.

**Part III – Minor Issues: Editorial and Data Presentation Modifications**

Reviewer #1: Please review the References page and fix the style.

The TMV work is an important complementation experiment showing the fusion protein functions and that the fusion is not malformed. I think this is important to state.

Reviewer #2: 1) Phase separation-deficient G3BP already restricted viral accumulation up to 5 folds, which is already very efficient. One interpretation of data is that phase separation, in this case, enhances the inhibitory role of G3BP in viral infection. G3BP has other intrinsic activity to sufficiently suppress viral accumulation.

2) Some rationales behind the experimental designs should be explained. For example, why particularly 1:6 molar ratio was used in line 224? There are numerous cases like this throughout the manuscript.

3) It is relevant to include a recent reference in discussion (Pubmed ID: 33910901).

Reviewer #3: Page 4 rows 84-88: PEMV 2 is a virus… in family Tombusviridae? Please, remind readers that taxonomically both PEMV 2 and GRV belong to genus Umbravirus of family Tombusviridae.

Page 6 row 130-131: it is stated that phase separation of IDR-GFP phase separation under crowding conditions could be observed by turbidity assay (Fig. 2D). Unfortunately, the quality of the Fig. 2D does not allow to see this.

Page 7 row 151: I don’t understand how the mean condensate sizes of all the other mutants are very similar except D/E-G. If I look the confocal image in 2E and 2G, I see differences. It would be good to explain which protein concentration was used to calculate this result.

Page 8 row 171: Please, explain what the basis to state is that the marked structures are nucleolus and Cajal bodies in the Fig. 3A. Did you use some markers here?

Page 12 row 285-188: The authors need to show that deltaNTF1-G3BP and G3BP are expressed on the same level (Fig. 5E) to make the conclusion that phase separation is needed for the full recovery of PEMV accumulation.

PLOS authors have the option to publish the peer review history of their article (what does this mean?). If published, this will include your full peer review and any attached files.

Reviewer #1: No

Reviewer #2: **Yes: **Ying Wang

Reviewer #3: No
---

## [Decision Letter · Decision Letter 1]

18 Aug 2021

Dear Dr. May,

Thank you very much for submitting your revised manuscript "Phase separation of both a plant virus movement protein and cellular factors support virus-host interactions" for consideration at PLOS Pathogens. We were able to secure two reviewers who reviewed your previous version. As you can see the reviews (below this email), reviewer 1 addressed substantial constructive comments for you to improve writing and we would like to invite the resubmission of a significantly-revised version that takes into account the reviewer 1's comments. You may contact the PLoS Pathogens editorial office to obtain the annotated pdf file from this reviewer. We are looking forward to receiving your new revision.

Sincerely,

Aiming Wang, Ph.D

Associate Editor

PLOS Pathogens

Shou-Wei Ding

Section Editor

PLOS Pathogens

Kasturi Haldar

Editor-in-Chief

PLOS Pathogens

orcid.org/0000-0001-5065-158X

Michael Malim

Editor-in-Chief

PLOS Pathogens

orcid.org/0000-0002-7699-2064

Reviewer's Responses to Questions

**Part I - Summary**

Reviewer #1: the study is novel, interesting, and significant. However the paper is very badly written. The M&M is very much of a mess. The results are circular arguments. In M&M i learned that a silencing suppressor was used throughout the study and so that should be explained in the results. The results need to elaborate experimental design, how data is collected and analyzed. how many replicates were performed. Statistics. Conclusions need to go into the discussion. The introduction and summary should highlight the viral basis for this study and stay away from neurodegenerative studies. the viral basis for this work is in the first paragraph of the discussion. I am providing an annotated pdf to help the authors rewrite. The cloning needs to be in one section of the M&M and PCR primers need to be explained and maybe put in a table. The M&M needs to provide sufficient information that others can repeat the work. Since the genes and mutants were synthesized the coding sequences need to be provided-otherwise no one can reproduce this work. gene accessions need to be provided. The discussion makes sweeping conclusions connecting in vitro data to systemic movement and this is a false argument.

Reviewer #2: The authors have fully addressed my concerns/comments. It could be beneficial to general readers if the authors include their explanation "PEMV2 will not tolerate addition of a fluorescent reporter" to methods or discussions during the publication process. Other than that, I recommend publishing this manuscript.

**Part II – Major Issues: Key Experiments Required for Acceptance**

Reviewer #1: The M&M indicates a silencing suppressor was included in most assays but this is not explained in the results and so I have questions about controls used or not used in the study. The results makes statements about measures and significance but does not state how measurements were performed, or statistics was performed.

Reviewer #2: None

**Part III – Minor Issues: Editorial and Data Presentation Modifications**

Reviewer #1: (No Response)

Reviewer #2: None

PLOS authors have the option to publish the peer review history of their article (what does this mean?). If published, this will include your full peer review and any attached files.

Reviewer #1: No

Reviewer #2: **Yes: **Ying Wang
---

## [Editor Report · Decision Letter 2]

13 Sep 2021

Dear Dr. May,

We are pleased to inform you that your manuscript 'Phase separation of a plant virus movement protein and cellular factors support virus-host interactions' has been provisionally accepted for publication in PLOS Pathogens.

Best regards,

Aiming Wang, Ph.D

Associate Editor

PLOS Pathogens

Shou-Wei Ding

Section Editor

PLOS Pathogens

Kasturi Haldar

Editor-in-Chief

PLOS Pathogens

orcid.org/0000-0001-5065-158X

Michael Malim

Editor-in-Chief

PLOS Pathogens

orcid.org/0000-0002-7699-2064

Thanks for properly addressing the comments from both reviewers, particularly Reviewer 1. This version reads very well and is acceptable for publication.
---

## [Editor Report · Acceptance letter]

15 Sep 2021

Dear Dr. May,

We are delighted to inform you that your manuscript, "Phase separation of a plant virus movement protein and cellular factors support virus-host interactions," has been formally accepted for publication in PLOS Pathogens.

Best regards,

Kasturi Haldar

Editor-in-Chief

PLOS Pathogens

orcid.org/0000-0001-5065-158X

Michael Malim

Editor-in-Chief

PLOS Pathogens

orcid.org/0000-0002-7699-2064